# GAUSS-FUSION: GAUSSIAN MEMORY AND CROSS-MODAL FUSION FOR 3D FEW-SHOT INCREMENTAL LEARNING

## ABSTRACT

Few-shot class-incremental learning (FSCIL) is a practical yet challenging problem in visual representation learning, primarily due to two notorious issues: catastrophic forgetting of previously learned classes and overfitting to the currently introduced classes. While most existing approaches adopt prototype-based mechanisms to address these challenges, achieving impressive results on 2D images that only a handful have explored FSCIL in the context of 3D point clouds. In this work, We propose GAUSS-Fusion, which combines (i) bidirectional cross-modal attention between 2D renderings and 3D points, (ii) a category-aware viewpoint selection strategy, and (iii) a Gaussian Memory for generative replay. Our method generates robust and homogeneous multi-modal representations. Extensive experiments demonstrate that our approach outperforms state-of-the-art approaches. Notably, our model achieves a reduction in the relative accuracy dropping rate $\Delta$ up to 10% on six benchmark 3D FSCIL tasks and two noisy-data tasks, showcasing its superior robustness and adaptability.

## 1 INTRODUCTION

Humans can continuously acquire new knowledge and strengthen their understanding without forgetting past knowledge. This capability is crucial in real-world scenarios, especially when new data samples are scarce, making the task even more challenging. This specific problem is known as Few-Shot Class-Incremental Learning (FSCIL). While there have been many FSCIL studies in the 2D domain, there is still significant room for expansion in the 3D domain. With the increasing availability of 3D point cloud data, we explore the possibility of implementing FSCIL in the 3D domain to address two primary challenges of FSCIL:

1. Catastrophic Forgetting:The model forgets knowledge of old classes while learning new ones.
2. Overfitting to a few new samples: The model overfits to the limited number of samples from the new classes.

Although there has been extensive research on the Few-Shot Class-Incremental Learning (FSCIL) problem for 2D images, the field remains relatively under-explored for 3D point cloud data.

Current 2D models can leverage large-scale vision-language models like CLIP and ALIGN, which are trained on vast amounts of data to achieve strong generalization to new classes. When faced with few-shot tasks, these models can effectively transfer their pre-trained knowledge to a small number of new classes, thereby avoiding overfitting to limited samples.

However, there exists a significant domain gap between 3D point cloud objects and the aforementioned 2D models. Bridging this gap is a crucial research topic. For example, the C3PR model attempts to address the FSCIL problem for 3D point clouds by incorporating the CLIP model. It aims to learn the optimal camera projection pose to project 3D point clouds onto 2D images.

To address the aforementioned challenges, we propose a novel multimodal prior framework. Since the spatial information of a 3D point cloud cannot be completely captured from a single viewpoint, our model does not rely on one best view. Instead, it utilizes the property that each category can

be represented by highly discriminative viewpoints, from which 2D visual features are generated to assist recognition. Furthermore, we improve the conventional max pooling operation in PointNet, which suffers from severe information loss, by introducing a bidirectional cross-modal attention mechanism. This mechanism utilizes the rich semantic clues of 2D images to guide the selection of local 3D geometric features, while simultaneously exploiting the global structural information of the 3D modality to reweight the importance of different 2D viewpoints. Therefore, our framework achieves mutual reinforcement across modalities.

During incremental learning, conventional methods often store a single sample for each old class to alleviate catastrophic forgetting. However, this approach is highly sensitive to sample selection. To overcome this limitation, we design a Gaussian Memory to record the statistical distribution (mean and standard deviation) of old class features. When learning new tasks, the model can sample from these distributions to replay prior knowledge. Our experiments confirm that this approach provides a more stable and effective solution to the forgetting problem.

In summary, the main contributions of our research are as follow:

1. **Bidirectional cross-modal attention** leverages the semantic clues of 2D images to guide the selection of local 3D geometric features(SGA), while simultaneously utilizing the global structural information from the 3D modality to reweight the importance of different 2D viewpoints(GGA).

2. **Dynamic viewpoint selection** to retain the most discriminative projections for the object.

3. **Gaussian Memory** that stores parametric feature distributions for replay, mitigating forgetting across sessions.

## 2 RELATED WORK

### 2.1 3D REPRESENTATION LEARNING FOR OBJECT RECOGNITION

With the rapid advancement of neural networks capable of learning high-quality representations, object recognition on 3D point clouds has achieved remarkable progress. As a pioneering approach, the PointNet series (Qi et al., 2017a;b) extracts both global and local features from point clouds using sampling, clustering, and projection techniques. By stacking multiple MLPs as a classification head, the PointNet series effectively predicts object categories from raw point clouds. Another prominent direction involves adapting convolutional modules originally designed for 2D images. For example, MVCNN (Su et al., 2015) aggregates features from multiview 2D renderings of a 3D object, enabling powerful recognition performance. DGCNN (Wang et al., 2019) introduces the edge convolution (EdgeConv) operator, which directly performs convolutions on the point cloud structure to capture local geometric relationships. More recently, hybrid approaches like PVCNN (Liu et al., 2019) leverage two branches: one branch voxelizes the point cloud and applies 3D convolution to obtain coarse-grained features, while the other branch directly processes the original point cloud to preserve fine-grained details. These features are then fused via devoxelization, yielding robust and expressive representations. Inspired by the success of attention-based architectures in 2D vision and language tasks, several methods, such as the Point Transformer series (Zhao et al., 2021; Wu et al., 2022; 2024), have extended transformers to point cloud data. Notably, the Point Transformer series replace multiplication with subtraction in the attention mechanism to reduce computational cost, while still achieving competitive performance. Finally, the integration of text features has emerged as a promising direction for learning 3D representation. Methods such as the PointCLIP series (Zhang et al., 2022; Zhu et al., 2023), CLIP2Point (Huang et al., 2023), Point-BERT (Yu et al., 2022), and PointGPT (Chen et al., 2023) leverage textual information, akin to CLIP (Radford et al., 2021) in 2D image domains, to learn more comprehensive and robust representations by aligning 3D objects with descriptive language. In this work, we adopt the multimodal encoder to extract homogeneous representations from multimodalities to obtain robustness and resilience.

### 2.2 FEW-SHOT CLASS-INCREMENTAL LEARNING

Continual and lifelong learning, especially within the class-incremental paradigm, emerge as pivotal research directions in recent years. Few-Shot Class-Incremental Learning (FSCIL), first formalized by Tao et al. (2020), represents a more realistic and challenging extension of this setting. A core

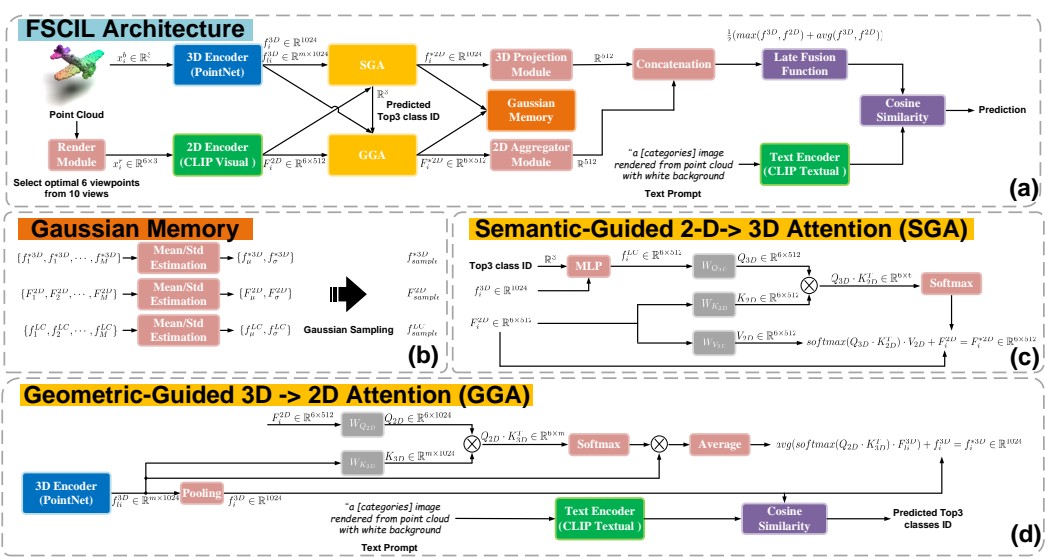

Figure 1: An overview of our proposed multi-modal architecture for 3D FSCIL. (a)The main architecture consists of 3D (PointNet) and 2D (CLIP) encoders whose outputs are refined and fused for the prediction. (b)The Gaussian Memory, which stores and samples statistical distributions of features from past classes to mitigate catastrophic forgetting. (c)The GGA, where a 3D-informed latent code refines multi-view 2D features. (d)The SGA, where 2D features guide the selection of local 3D point features.

motivation for FSCIL stems from the prevalence of synthetic data which is far more abundant than real-world data enabling the pre-training of models on large-scale synthetic datasets, followed by rapid adaptation to novel target domains with only a handful of labeled examples. This challenge is particularly acute in 3D domains (*e.g.*, outdoor driving datasets), where annotated data is even scarcer than in traditional 2D image tasks. In addition to the well-known issue of catastrophic forgetting in incremental learning, FSCIL introduces further complications, such as pronounced overfitting to newly introduced classes. Several recent works (Cheraghian et al., 2021a; Mazumder et al., 2021; Cheraghian et al., 2021b; Yang et al., 2023) address these challenges by leveraging class prototypes for base and novel classes separately, achieving promising results and providing valuable inspiration for advancing FSCIL in the 3D regime. However, compared to its 2D counterpart, 3D FSCIL remains significantly underexplored.

To bridge this gap, Chowdhury et al. (2022) introduce the notion of "micro-shape" components, replacing holistic class prototypes with collections of part-based representations (*e.g.*, leg, plane, back). Their approach mines these fundamental components during base training and augments their learned features with word2vec (Mikolov et al., 2013) embeddings to mitigate both forgetting and overfitting in later sessions, ultimately improving inference-time performance. More recently, Cheraghian et al. (2024) integrate vision-language (VL) models into the FSCIL pipeline, rendering images from 3D point clouds and enforcing alignment among 3-D, image, and text features. By adopting a gradient-based rendering model, their approach enables end-to-end optimization of the projection matrix, yielding more informative and view-optimized rendered images. While the main scope focuses on incremental learning, Tian et al. (2025) extend the part-concept paradigm by introducing a codebook of part components, further demonstrating the effectiveness of part-based representations through additional FSCIL experiments. Building on these insights, our approach introduces multimodal priors by analogy to part-concept learning, combining optimally rendered images and class name text embeddings to fuse diverse perspectives. This enables the construction of more robust and homogeneous representations, thereby enhancing both generalization and resilience in few-shot 3D incremental learning.

## 3 METHOD

Our proposed framework, *GAUSS-Fusion*, integrates geometric and semantic information across modalities while preserving past knowledge for incremental learning. As illustrated in Figure 1(a), GAUSS-Fusion is composed of four main components: a bidirectional cross-modal attention mechanism (SGA and GGA), a dynamic viewpoint selection strategy, a *Gaussian Memory* for generative replay, and a late-fusion classifier that aligns multi-modal features with text-based prompts.

### 3.1 PROBLEM FORMULATION

In our FSCIL setting, the dataset for each task $t$ is denoted as $D^t = \{(\mathcal{X}_i^t, \mathbf{y}_i^t, \mathbf{p}_i^t)\}_{i=1}^{n_t}$, where $\mathcal{X}_i^t$ is a 3D point cloud sample with coordinates $\mathbf{x}_i^t \in \mathbb{R}^3$, $n_t$ is the number of samples in task $t$, and $\mathbf{y}_i^t \in \mathcal{Y}^t$ together with $\mathbf{p}_i^t \in \mathcal{P}^t$ represent prompt-based class descriptions. The base task ($t = 0$) provides a large-scale training dataset, while tasks with $t > 0$ correspond to incremental few-shot tasks where each class contains only a few samples. The model is trained sequentially over tasks $t = 0, \ldots, T$. At task $t$, it has access to $\mathcal{X}^t$, $\mathcal{Y}^t$, and all accumulated prompts $\{\mathcal{P}^0, \ldots, \mathcal{P}^t\}$. During inference, the model must recognize test samples from both current and all previously learned tasks.

### 3.2 SEMANTIC-GUIDED 2D→3D ATTENTION (SGA)

PointNet Qi et al. (2017a) aggregates point cloud features through symmetric pooling, typically max pooling. Although this guarantees permutation invariance, it discards much of the geometric signal: only the maximum activation per dimension is retained, while other informative contributions are lost. As a result, fine-grained cues such as edges, corners, and local structural variations are suppressed, producing global representations that are less discriminative and insufficiently expressive for few-shot incremental learning.

To overcome this limitation, we propose a *semantic-guided attention* (SGA) mechanism that leverages semantic cues from the 2D modality to refine 3D feature aggregation. In this design, 2D features $F_i^{2D}$ extracted from CLIP act as queries that emphasize discriminative 3D points (Figure 1(d)). For a point cloud sample $\mathcal{X}_i^t$, PointNet generates per-point features $F_{\ell i}^{3D}$, preserving local geometry. In parallel, the same point cloud is rendered into depth images from selected viewpoints and processed by CLIP to yield semantic features $F_i^{2D}$. We express the cross-modal attention module as follows:

$$f_i^{*3D} = \text{avg}\left(\text{softmax}(Q_{2D}K_{3D}^\top) \cdot F_{\ell i}^{3D}\right) + f_i^{3D}, \tag{1}$$

where $Q_{2D}$ is derived from $F_i^{2D}$ and $K_{3D}$ corresponds to local 3D features. The SGA mechanism allows semantic cues to highlight contours, surfaces, and distinctive geometric parts most relevant to category-level discrimination. Unlike max pooling, which treats all points uniformly, SGA adaptively emphasizes informative subsets of the point cloud, yielding robust global 3D representations well suited for incremental classification.

### 3.3 GEOMETRIC-GUIDED 3D→2D ATTENTION (GGA)

To establish a complementary bidirectional flow, we design a *geometric-guided attention* (GGA) mechanism that refines multi-view 2D features using structural cues from the 3D domain (Figure 1(c)). Based on the viewpoint selection strategy described in Section 3.4, each 3D sample is projected into multiple 2D features $F_i^{2D}$. However, not all views are equally informative; some may suffer from occlusion or capture the object from unfavorable angles that obscure discriminative details. To overcome this, GGA integrates two sources of 3D information: (1) the global 3D feature $f_i^{3D}$, which encodes the object's geometry, and (2) the top-$M$ candidate categories predicted by PointNet. (We have used $M = 3$ in all our experiments.)

These features are processed by an MLP to generate a latent code $f_i^{LC}$, which serves as the query $Q_{3D}$. The multi-view image features act as keys and values ($K_{2D}, V_{2D}$), and attention weights are computed as

$$F_i^{*2D} = \text{softmax}(Q_{3D}K_{2D}^\top) \cdot V_{2D} + F_i^{2D}. \tag{2}$$

This process enables the 3D representation to highlight the most discriminative viewpoints, producing refined 2D features that complement the 3D stream and enhance classification.

## 3.4 VIEWPOINT SELECTION STRATEGY

To exploit the image–text alignment capability of CLIP, 3D point clouds are projected into 2D depth images. Since viewpoint choice significantly affects recognition, and optimal views vary across categories, we adopt a category-aware viewpoint selection strategy. Before training, we build a view library for each category by rendering all training samples from ten candidate viewpoints. Each view is evaluated with CLIP using cross-entropy loss, and the top $k$ views per category are retained as representative viewpoints.

During training and inference, a PointNet backbone first predicts the $M$ most likely candidate categories for a given sample. For each candidate, we retrieve its top $k_v$ views from the library, forming a projection set of $M \times k_v$ views. This strategy improves efficiency and ensures that rendered depth images capture the most discriminative perspectives.

## 3.5 FUSION-BASED CLASSIFICATION

Refined 3D and 2D features are projected into the same dimension by a 3D projection module and a 2D aggregator, then fused using

$$f^{fusion} = \tfrac{1}{2}\left(\max(f^{3D}, f^{2D}) + \mathrm{avg}(f^{3D}, f^{2D})\right). \tag{3}$$

The fused feature is aligned with text embeddings of class descriptions from CLIP, and classification is performed by cosine similarity, with the highest-scoring class selected as the prediction.

## 3.6 GAUSSIAN MEMORY FOR INCREMENTAL LEARNING

Incremental learning suffers from catastrophic forgetting because parameters are optimized for the current task while prior knowledge is overwritten. To mitigate this, we propose a *Gaussian Memory* that stores compact parametric distributions of past classes (Figure 1(b)).

After training each task, we store the mean and variance of three feature types per class: (1) global 3D features $\left(f_\mu^{3D}, f_\sigma^{3D}\right)$, (2) aggregated 2D features $\left(F_\mu^{2D}, F_\sigma^{2D}\right)$, and (3) latent codes $\left(f_\mu^{LC}, f_\sigma^{LC}\right)$. In subsequent tasks, we sample from these Gaussians to replay pseudo-features of past classes, combining them with real samples $\{\mathcal{X}_i^t\}_{i=1}^{n_t}$ from the current task. This generative replay strategy alleviates forgetting while enabling the model to continuously adapt to new classes.

## 4 EXPERIMENTS

### 4.1 DATASET

We evaluate the performance of our framework following the FSCIL dataset settings proposed in FSCIL-3D. A total of two real-world 3D datasets, ScanObjectNN and CO3D, and two synthetic 3D datasets, ModelNet40 and ShapeNet, were used. Each 3D point cloud contains 1024 points. Leveraging these datasets, we designed two FSCIL scenarios: within-dataset and cross-dataset. The within-dataset scenario is relatively straightforward, where both base and novel tasks originate from the same dataset. In contrast, the cross-dataset scenario is more complex and practical, where the model is trained on synthetic data for the base tasks and evaluated on novel tasks from the real world, few-shot datasets.

### 4.2 IMPLEMENTAION DETAILS

We use PointNet as the model for 3D point clouds, leveraging the vision encoder $(V_e)$ and text encoder $(T_e)$ from the CLIP ViT-B/16 architecture.

First, we pretrain the PointNet using the base tasks. Subsequently, we train our model using the AdamW optimizer and CosineAnnealingLR as the scheduler. During training, we apply data augmentation techniques such as random translation, scaling of the input point cloud, and random point dropout to enhance the model's robustness. For the base tasks, the learning rate is set to 0.00002, and we train for a total of 200 epochs. For the novel class tasks, we also use the AdamW optimizer, fine-tuning the model for 50 epochs with a learning rate of 0.001 for CO3D and ModelNet, and 0.0002 for the other datasets.

Table 1: FSCIL results for within-dataset experiment

| Method | ShapeNet (Chang et al., 2015) | | | | | | | | ModelNet (Wu et al., 2015) | | | | | | CO3D (Reizenstein et al., 2021) | | | | | | |
|---|---|---|---|---|---|---|---|---|---|---|---|---|---|---|---|---|---|---|---|---|---|
| | 25 | 30 | 35 | 40 | 45 | 50 | 55 | $\Delta\downarrow$ | 20 | 25 | 30 | 35 | 40 | $\Delta\downarrow$ | 25 | 30 | 35 | 40 | 45 | 50 | $\Delta\downarrow$ |
| FT | 87.0 | 25.7 | 6.8 | 1.3 | 0.9 | 0.6 | 0.4 | 99.5 | 89.8 | 9.7 | 4.3 | 3.3 | 3.0 | 96.7 | 76.7 | 11.2 | 3.6 | 3.2 | 1.8 | 0.8 | 99.0 |
| Joint | 87.0 | 85.2 | 84.3 | 83.0 | 82.5 | 82.2 | 81.3 | 6.6 | 89.8 | 88.2 | 87.0 | 83.5 | 80.5 | 10.4 | 76.7 | 69.4 | 64.8 | 62.7 | 60.7 | 59.8 | 22.0 |
| EEIL (Castro et al., 2018) | 87.0 | 77.7 | 73.2 | 69.3 | 66.4 | 65.9 | 65.8 | 22.4 | 89.8 | 75.4 | 67.2 | 60.1 | 55.6 | 38.1 | 76.7 | 61.4 | 52.4 | 42.8 | 39.5 | 32.8 | 57.2 |
| FACT (Zhou et al., 2022) | 87.5 | 75.3 | 71.4 | 69.9 | 67.5 | 65.7 | 62.5 | 28.6 | 90.4 | 81.3 | 77.1 | 73.5 | 65.0 | 28.1 | 77.9 | 67.1 | 59.7 | 54.8 | 50.2 | 46.7 | 40.0 |
| Sem-aware (Cheraghian et al., 2021a) | 87.2 | 74.9 | 68.1 | 69.0 | 68.1 | 66.9 | 63.8 | 26.8 | 91.3 | 82.2 | 74.3 | 70.0 | 64.7 | 29.1 | 76.8 | 66.9 | 59.2 | 53.6 | 49.1 | 42.9 | 44.1 |
| FSCIL-3D (Chowdhury et al., 2022) | 87.6 | 83.2 | 81.5 | 79.0 | 76.8 | 73.5 | 72.6 | 17.1 | 93.6 | 83.1 | 78.2 | 75.8 | 67.1 | 28.3 | 78.5 | 67.3 | 60.1 | 56.1 | 51.4 | 47.2 | 39.9 |
| C3PR (Cheraghian et al., 2024) | 88.0 | 81.6 | 77.8 | 76.7 | 76.9 | 76.2 | 74.7 | 15.1 | 91.6 | 82.3 | 75.8 | 72.2 | 70.9 | 22.5 | 81.5 | 69.4 | 66.5 | 63.0 | 54.2 | 53.8 | 34.0 |
| **Ours** | **88.8** | **86.8** | **85.4** | **84.8** | **84.2** | **83.7** | **83.6** | **5.9** | **93.6** | **87.6** | **86.3** | **83.0** | **81.8** | **12.6** | **80.8** | **74.6** | **71.8** | **71.4** | **68.2** | **67.0** | **17.0** |

Following FSCIL-3D, we also calculate the relative accuracy dropping rate, $\Delta = \frac{|acc_T - acc_0|}{acc_0} \times 100$ where $acc_T$ and $acc_0$ represent the accuracy of the last and first incremental tasks, respectively. A lower $\Delta$ value indicates better overall performance of the method.

## 4.3 MAIN RESULT

### 4.3.1 COMPARED METHODS

To objectively assess the effect of our proposed method, we adopt several FSCIL methods for comparison, following the protocol of C3PR (Cheraghian et al., 2024). These include EEIL (Castro et al., 2018), LwF, FACT (Zhou et al., 2022), Sem-aware (Cheraghian et al., 2021a), FSCIL-3D (Chowdhury et al., 2022), and C3PR (Cheraghian et al., 2024). It is important to note that among these methods, only FSCIL-3D (Chowdhury et al., 2022) and C3PR (Cheraghian et al., 2024) are specifically designed for point cloud data, while the others were originally developed for image-based tasks. Despite the difference in the data setting for which these methods were created, such comparisons remain an informative means of estimating the relative performance of our approach across different datasets.

### 4.3.2 WITHIN-DATASET

To validate the effectiveness of the proposed multimodal framework, we conduct within-dataset incremental learning experiments on three datasets according to FSCIL-3D: ShapeNet, ModelNet, and CO3D. In this setting, approximately half of the categories are designated as the base task, while the remaining categories are input in a few-shot mode across incremental stages. The model is therefore required to acquire new categories while simultaneously preserving its recognition performance on previously learned categories.

The experimental results, summarized in Table 1, indicate our approach consistently outperforms the above methods across all within-datasets. Especially, with respect to the metric of relative accuracy dropping rate ($\Delta$), our method achieves the lowest values of 5.9, 12.6, and 17.0 on ShapeNet, ModelNet, and CO3D, respectively, surpassing the other methods by a large margin. These experiments provide clear evidence that the proposed cross-modal attention mechanism and the Gaussian Memory strategy substantially mitigate the problem of catastrophic forgetting, hence enabling the model to retain high recognition accuracy on old categories while continually acquiring new knowledge.

### 4.3.3 CROSS-DATASET

To further demonstrate the robustness and generalization of our method in actual applications, we conduct harder cross-dataset incremental learning experiments, following the FSCIL-3D protocol. These experiments are desined to imitate the domain shift from ideal synthetic data to complex real-world data. In this setting, the base task is trained on clean synthetic datasets (ShapeNet or ModelNet40), while the model is required to learn novel categories from real-scanned datasets (CO3D or ScanObjectNN) under few-shot conditions. Real-world datasets often contain missing parts and noise, making this scenario substantially more challenging than the within-dataset experiments.

Within this framework, we perform comprehensive evaluations across three scenarios: ShapeNet → CO3D, ModelNet40 → ScanObjectNN, and ShapeNet → ScanObjectNN. The results, summarized in Table 2, show that our method consistently and significantly outperforms other compared approaches across every setting. In particular, with respect to the metric of relative accuracy dropping rate ($\Delta$), our method achieves the lowest values of 18.5, 17.7, and 6.5, respectively, superior

Table 2: FSCIL results for cross-dataset experiment

| Method | ShapeNet → CO3D | | | | | | | | | | | | ModelNet → ScanObjectNN | | | | | ShapeNet → ScanObjectNN | | | | |
|---|---|---|---|---|---|---|---|---|---|---|---|---|---|---|---|---|---|---|---|---|---|---|
| | 39 | 44 | 49 | 54 | 59 | 64 | 69 | 74 | 79 | 84 | 89 | Δ↓ | 26 | 30 | 34 | 37 | Δ↓ | 44 | 49 | 54 | 59 | Δ↓ |
| FT | 81.0 | 20.2 | 2.3 | 1.7 | 0.8 | 1.0 | 1.0 | 1.3 | 0.9 | 0.5 | 1.6 | 98.0 | 88.4 | 6.4 | 6.0 | 1.9 | 97.9 | 81.4 | 38.7 | 4.0 | 0.9 | 98.9 |
| Joint | 81.0 | 79.5 | 78.3 | 75.2 | 75.1 | 74.8 | 72.3 | 71.3 | 70.0 | 68.8 | 67.3 | 16.9 | 88.4 | 79.7 | 74.0 | 71.2 | 19.5 | 81.4 | 82.5 | 79.8 | 78.7 | 3.3 |
| EEIL (Castro et al., 2018) | 81.0 | 75.2 | 69.3 | 63.2 | 60.5 | 57.9 | 53.0 | 51.9 | 51.3 | 47.8 | 47.6 | 41.2 | 88.4 | 70.2 | 61.0 | 56.8 | 35.7 | 81.4 | 74.5 | 69.8 | 63.4 | 22.1 |
| FACT (Zhou et al., 2022) | 81.0 | 76.0 | 70.3 | 68.1 | 65.8 | 63.5 | 63.0 | 60.1 | 58.2 | 57.5 | 55.9 | 31.3 | 89.1 | 72.5 | 68.3 | 63.5 | 28.7 | 82.3 | 74.6 | 69.9 | 66.8 | 18.8 |
| Sem-aware (Cheraghian et al., 2021a) | 81.4 | 69.5 | 66.5 | 62.9 | 63.2 | 63.0 | 61.2 | 58.3 | 58.1 | 57.2 | 55.2 | 31.6 | 88.5 | 73.9 | 67.7 | 64.2 | 27.5 | 81.3 | 70.6 | 65.2 | 62.9 | 22.6 |
| FSCIL-3D (Chowdhury et al., 2022) | 82.6 | 77.9 | 73.9 | 72.7 | 67.7 | 66.2 | 65.4 | 63.4 | 60.6 | 58.1 | 57.1 | 30.9 | 89.3 | 73.2 | 68.4 | 65.1 | 27.1 | 82.5 | 74.8 | 71.2 | 67.1 | 18.7 |
| C3PR (Cheraghian et al., 2024) | 83.6 | 80.0 | 77.8 | 75.4 | 72.8 | 72.3 | 70.3 | 67.9 | 64.9 | 64.1 | 63.2 | 24.4 | 88.3 | 75.7 | 70.6 | 67.8 | 23.2 | 84.5 | 77.8 | 75.5 | 71.9 | 14.9 |
| **Ours** | **85.5** | **84.0** | **82.0** | **79.4** | **78.3** | **76.4** | **75.0** | **73.5** | **72.7** | **71.2** | **69.7** | **18.5** | **90.8** | **82.0** | **77.6** | **73.9** | **17.7** | **87.2** | **84.9** | **83.3** | **81.5** | **6.5** |

to other methods. These experiments prove that the proposed framework effectively handles the difficulties of domain gaps by superior knowledge retention, even when learning with real-world data distributions that differ substantially from the base task.

## 4.4 NOISE IMMUNITY ANALYSIS

To evaluate the robustness of our proposed method under noisy conditions, we design a noise-resilience experiment to simulate the point clouds influenced by sensors or environmental factors. In this experiment, we apply point shift noise to the point clouds by adding an independently sampled perturbation vector to each point's coordinates, drawn from a zero-mean Gaussian distribution $N(0, \sigma^2 I)$. We consider two levels of noise intensity: a low-intensity setting with $\sigma = 0.001$ and a high-intensity setting with $\sigma = 0.01$ to inspect the stability of the model.

The results of the experiment are summarized in Table 3. It should be noted that, because of differences in experimental configurations, the baseline models reproduced from the official FSCIL-3D code appear to achieve better $\Delta$ values under low-intensity noise compared to their original noiseless results. Therefore, the focus of this analysis is not on the absolute values, but rather on the extent of performance degradation when transitioning from low-intensity to high-intensity noise.

The results demonstrate the superior stability of our approach. On the within-dataset setting (CO3D), when the noise intensity increases from $\sigma = 0.0001$ to $\sigma = 0.001$ the $\Delta$ of our method increases by only 1.9 (from 18.2 to 20.1), while FSCIL-3D exhibits a larger increase of 2.9 (from 34.5 to 37.4). Similarly, under the cross-dataset setting (ModelNet → ScanObjectNN), our method shows a $\Delta$ rise of merely 1.2 (from 19.1 to 20.3), compared to 2.0 (from 25.2 to 27.2) for FSCIL-3D. These results clearly indicate that the proposed multimodal framework possesses strong noise-resilience.

## 4.5 ROBUSTNESS TO RANDOM DROP

To further investigate the robustness of our model, we simulate the data incompleteness problem commonly encountered in the real world due to object occlusion or incomplete scans. In this experiment, we apply a random point drop to the input point clouds, with two levels of interference: a moderate missing rate with a max drop rate = 0.25 and a severe missing rate with a max drop rate = 0.5.

The experimental results are reported in Table 4. Similar to the noise-resilience analysis presented earlier, our focus here is on the extent of performance degradation across different levels of dropping. The results disclose that our method exhibits superior stability. On the within-dataset setting (CO3D), when the max drop rate increases from 25% to 50%, the $\Delta$ metric of our approach rises by only 4.9 (from 18.2 to 23.1), which is substantially smaller than the 6.7 increase observed in the FSCIL-3D (from 30.0 to 36.7). This robustness is even more pronounced in the cross-dataset setting (ModelNet → ScanObjectNN), where our model maintains the same $\Delta$ of 19.7 across both drop rates, showing virtually no performance degradation, whereas FSCIL-3D continues to decline.

## 4.6 ROBUSTNESS IN EXTREME FEW-SHOT SCENARIOS

To validate our model's robustness in extreme few-shot scenarios, we designed an experiment that decreases the number of available samples for new classes in the incremental stage, from a 5-shot down to the most challenging 1-shot setting. The experimental results (Figure 2) reveal a significant advantage of our method: its performance with only a single sample (1-shot) surpasses the 5-shot

Table 3: Performance comparison under different levels of Gaussian point shift noise.

| Method | CO3D | | | | | | | ModelNet → ScanObjectNN | | | | |
|---|---|---|---|---|---|---|---|---|---|---|---|---|
| | 25 | 30 | 35 | 40 | 45 | 50 | $\Delta\downarrow$ | 26 | 30 | 34 | 37 | $\Delta\downarrow$ |
| FSCIL-3D (Chowdhury et al., 2022) | 78.5 | 67.3 | 60.1 | 56.1 | 51.4 | 47.2 | 39.9 | 89.3 | 73.2 | 68.4 | 65.1 | 27.1 |
| FSCIL-3D* (noise $\sigma$=0.001) | 78.1 | 69.3 | 64.6 | 61.5 | 52.6 | 51.1 | 34.5 | 86.5 | 74.9 | 68.6 | 64.7 | 25.2 |
| FSCIL-3D* (noise $\sigma$=0.001) | 77.5 | 67.3 | 60.6 | 56.5 | 51.9 | 48.5 | 37.4 | 86.3 | 74.6 | 66.8 | 62.8 | 27.2 |
| **Ours** | **80.8** | **74.6** | **71.8** | **71.4** | **68.2** | **67.0** | **17.0** | **90.8** | **82.0** | **77.6** | **73.9** | **17.7** |
| **Ours(noise $\sigma$=0.001)** | **80.8** | **72.8** | **70.3** | **68.9** | **66.4** | **66.1** | **18.2** | **90.5** | **81.0** | **75.4** | **73.2** | **19.1** |
| **Ours(noise $\sigma$=0.01)** | **79.9** | **70.7** | **67.9** | **66.5** | **64.1** | **63.8** | **20.1** | **89.7** | **79.6** | **74.0** | **71.5** | **20.3** |

* : Our experiments are based on the official implementation of FSCIL-3D.

Table 4: Robustness analysis under random point dropping

| Method | CO3D | | | | | | | ModelNet → ScanObjectNN | | | | |
|---|---|---|---|---|---|---|---|---|---|---|---|---|
| | 25 | 30 | 35 | 40 | 45 | 50 | $\Delta\downarrow$ | 26 | 30 | 34 | 37 | $\Delta\downarrow$ |
| FSCIL-3D (Chowdhury et al., 2022) | 78.5 | 67.3 | 60.1 | 56.1 | 51.4 | 47.2 | 39.9 | 89.3 | 73.2 | 68.4 | 65.1 | 27.1 |
| FSCIL-3D* (max drop rate=0.25) | 73.9 | 65.8 | 61.5 | 58.2 | 52.0 | 51.7 | 30.0 | 80.9 | 70.8 | 61.5 | 58.7 | 27.4 |
| FSCIL-3D* (max drop rate=0.5) | 63.6 | 57.2 | 53.0 | 49.8 | 44.5 | 40.4 | 36.7 | 68.0 | 60.1 | 51.6 | 49.1 | 27.8 |
| **Ours** | **80.8** | **74.6** | **71.8** | **71.4** | **68.2** | **67.0** | **17.0** | **90.8** | **82.0** | **77.6** | **73.9** | **17.7** |
| **Ours (max drop rate=0.25)** | **80.8** | **72.8** | **70.3** | **68.9** | **66.4** | **66.1** | **18.2** | **90.4** | **82.0** | **77.6** | **73.9** | **19.7** |
| **Ours (max drop rate=0.5)** | **76.9** | **66.0** | **62.7** | **61.4** | **59.4** | **59.1** | **23.1** | **89.0** | **79.3** | **74.2** | **71.5** | **19.7** |

* : Our experiments are based on the official implementation of FSCIL-3D.

benchmark of competing methods. This demonstrates the model's superior generalization capability and resistance to overfitting.

## 4.7 ABLATION STUDY

To gain deeper insights into the specific contributions of each key module in our model, we conduct a series of detailed ablation studies. By selectively removing or simplifying each component, we can quantitatively assess the practical influence of each module on performance. All experimental results are summarized in Table 5.

### 4.7.1 IMPACT OF VIEWPOINT SELECTION STRATEGY

First, we assess the impact of the viewpoint selection strategy. We design two variants: (1) removing the dynamic selection mechanism and instead assigning all objects a fixed pair of viewpoints (ours (2 views)); and (2) retaining the dynamic selection process but reducing the number of viewpoints per candidate category from two to a single best viewpoint (ours (3 cls1view)).

The experiments show that the fixed two-view configuration suffers the most significant performance degradation, with the $\Delta$ deteriorating to 20.5 on CO3D and 20.0 on ModelNet → ScanObjectNN. The single-view configuration performs slightly better, with $\Delta$ values of 18.6 and 18.3, respectively, but still worse than our full model, which achieves 17.0 and 17.7. These findings confirm that dynamic and diverse viewpoint selection is crucial for generating high-quality two-dimensional features.

### 4.7.2 EFFECTIVENESS OF BIDIRECTIONAL ATTENTION

Next, we evaluate the effectiveness of the bidirectional cross-modal attention mechanism (SGA and GGA). When all attention modules are removed (ours (without Attention)), the two-dimensional and three-dimensional features are no longer mutually weighted or refined.

The experimental results show that this modification leads to an increase in the $\Delta$ metric, reaching 21.1 on CO3D and 19.2 on ModelNet → ScanObjectNN. These results clearly demonstrate that the mutual refinement between cross-modal features is crucial for generating more discriminative multi-modal representations.

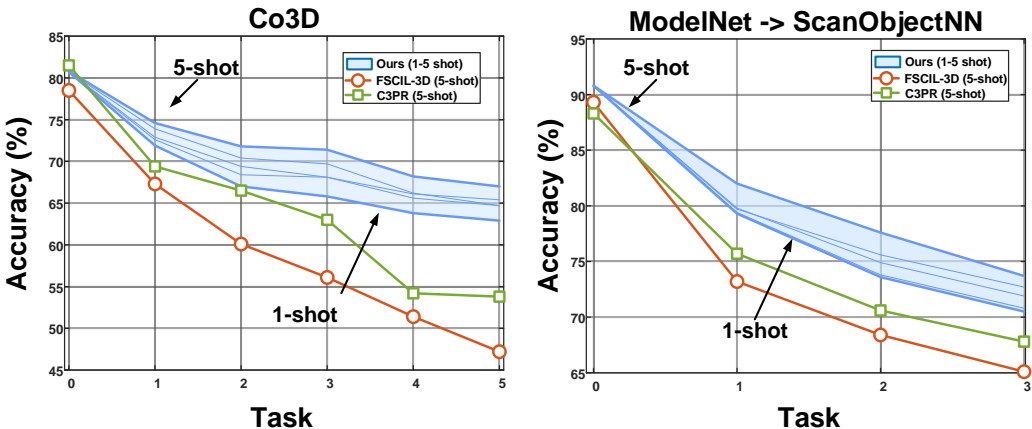

Figure 2: Performance comparison under different few-shot scenarios.

Table 5: Ablation analysis of key components in our model

| Method | CO3D (Reizenstein et al., 2021) | | | | | | | ModelNet $\rightarrow$ ScanObjectNN | | | | |
|---|---|---|---|---|---|---|---|---|---|---|---|---|
| | 25 | 30 | 35 | 40 | 45 | 50 | $\Delta \downarrow$ | 26 | 30 | 34 | 37 | $\Delta \downarrow$ |
| **Ours** | **80.8** | **74.6** | **71.8** | **71.4** | **68.2** | **67.0** | **17.0** | **90.8** | **82.0** | **77.6** | **73.9** | **17.7** |
| ours (3 cls1view) | 80.8 | 72.5 | 69.7 | 69.2 | 66.6 | 65.8 | 18.6 | 90.8 | 81.1 | 76.5 | 73.2 | 18.3 |
| ours (2 views) | 80.8 | 72.0 | 68.7 | 68.1 | 66.0 | 64.4 | 20.5 | 90.8 | 80.0 | 75.8 | 72.9 | 20.0 |
| ours (without Attetion) | 80.4 | 70.7 | 67.8 | 66.4 | 65.1 | 63.4 | 21.1 | 90.8 | 81.2 | 76.2 | 73.4 | 19.2 |
| ours (without Gaussian Memory) | 80.8 | 72.5 | 68.2 | 65.6 | 62.8 | 61.7 | 23.6 | 90.8 | 81.2 | 73.7 | 70.6 | 22.3 |

### 4.7.3 CONTRIBUTION OF GAUSSIAN MEMORY

Finally, we verify the critical role of the Gaussian Memory module in mitigating catastrophic forgetting. In this experiment, we completely remove the module (ours (without Gaussian Memory)). As FSCIL-3D, the model randomly samples one instance from a previously seen category.

As shown in Table 5, the absence of Gaussian Memory results in a significant performance decline: the $\Delta$ metric increases sharply from 17.0 to 23.6 on CO3D, and from 17.7 to 22.3 on ModelNet $\rightarrow$ ScanObjectNN. These results strongly highlight that leveraging stored feature distributions for knowledge prevents rapid performance degradation.

## 5 CONCLUSIONS

This research aims to address the core challenges of FSCIL in the 3D point cloud domain, namely catastrophic forgetting and overfitting. To this end, we propose an innovative gauss-fusion framework that integrates point clouds, multi-view rendered images, and textual information to construct a more comprehensive representation. We design a bidirectional cross-modal attention mechanism, in which the rich semantics of 2D features guide the selection of 3D geometric features (SGA). In contrast, the global 3D geometry reweights the importance of different 2D viewpoints (GGA), hence achieving mutual enhancement across modalities.

To mitigate catastrophic forgetting, we introduce the Gaussian Memory, which stores and samples the statistical distributions of old category features rather than unstable individual instances, thereby achieving effective knowledge consolidation. Extensive experiments across multiple datasets demonstrate that our method achieves state-of-the-art performance in both within-dataset and more challenging cross-dataset and noisy data settings. In particular, with respect to the critical metric of the relative accuracy dropping rate $\Delta$, our model consistently shows substantial advantages, hence validating its superior robustness and generalization.

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

## A APPENDIX

### A.1 USING AN LLM TO HELP WITH PAPER WRITING

During the preparation of this manuscript, we utilized Large Language Models (LLMs) as assistive tools. Specifically, Google Gemini and ChatGPT were used to improve grammar, polish phrasing, and assist in drafting the initial structure of the paper. All final content was carefully reviewed, revised, and confirmed by the authors, who bear full responsibility for the entirety of this work.

