# OpenReview forum: "GAUSS-Fusion: Gaussian Memory and Cross-Modal Fusion for 3D Few-Shot Incremental Learning"
_ICLR.cc/2026/Conference — ICLR 2026 Conference Withdrawn Submission_

### Official Review · Reviewer_BXaq · 2025-10-29

**Soundness:** 2
**Presentation:** 2
**Contribution:** 2
**Rating:** 4
**Confidence:** 3

**Summary:**

This paper proposes GAUSS-Fusion, a framework for Few-Shot Class-Incremental Learning (FSCIL) in 3D point clouds. The method incorporates bidirectional cross-modal attention between 2D images and 3D point clouds, a category-aware viewpoint selection strategy, and a Gaussian Memory for generative replay to mitigate catastrophic forgetting. The framework is designed to handle the challenges of catastrophic forgetting and overfitting in the context of FSCIL, providing robust and adaptable performance across both synthetic and real-world datasets.

**Strengths:**

1. The use of bidirectional cross-modal attention to combine 2D and 3D features makes sense to me. Gaussian Memory is effective for mitigating catastrophic forgetting in FSCIL for 3D point clouds.
2. The experiment on multiple few-shot scenarios is interesting, demonstrating the effectiveness of the proposed method in few-shot tasks.

**Weaknesses:**

1. GAUSS-Fusion uses the pre-trained CLIP model for feature extraction, which raises concerns about whether the performance improvements are primarily due to CLIP or the newly proposed method. The results of the ablation study in Table 5 also show that the baseline performance is already higher than the compared methods.
2. The experimental comparisons are not exhaustive enough. For example, FILP3D [1], which also combines 2D and 3D features like the proposed method, is not included in the comparison.
3. Gaussian memory is widely used [2,3] in continuous learning methods, which makes its use here lack significant technical novelty.

[1] FILP-3D: Enhancing 3D few-shot class-incremental learning with pre-trained vision-language models. Pattern Recognition, 2025, 165: 111558.
[2] Slca: Slow learner with classifier alignment for continual learning on a pre-trained model. Proceedings of the IEEE/CVF International Conference on Computer Vision. 2023: 19148-19158.
[3] Few-shot class incremental learning leveraging self-supervised features. Proceedings of the IEEE/CVF conference on computer vision and pattern recognition. 2022: 3900-3910.

**Questions:**

Please refer to the weaknesses above. If the authors can address my concerns, I would be willing to increase my score.

---

### Official Review · Reviewer_XYwW · 2025-10-30

**Soundness:** 2
**Presentation:** 2
**Contribution:** 2
**Rating:** 2
**Confidence:** 3

**Summary:**

This paper addresses the challenge of few-shot class-incremental learning (3D FSCIL) in the domain of 3D point clouds by leveraging the strong generalization of the 2D pre-trained CLIP model to effectively transfer knowledge in few-shot learning scenarios. However, there exists a significant domain gap between 3D point clouds and 2D image models. To bridge this gap, the authors propose a bidirectional cross-modal attention mechanism and a category-aware dynamic viewpoint selection strategy. Furthermore, to tackle the issue of catastrophic forgetting in incremental learning, the authors introduce a Gaussian Memory module that stores the statistical distributions of features and replays sampled features to prevent catastrophic forgetting. The experimental results presented in the paper demonstrate that the proposed method outperforms the previous methods; however, it lacks a detailed analysis of the reasons behind the performance improvements.

**Strengths:**

1. The experimental results presented in the paper demonstrate that the proposed method outperforms the previous methods; however, it lacks a detailed analysis of the reasons behind the performance improvements.

**Weaknesses:**

1. The paper lacks a clear analysis of the problem and motivation. In both the abstract and the introduction, the authors fail to provide a sufficient discussion of the limitations of previous methods or the motivation for their proposed solution. Both the proposed method and the C3PR [1] approach incorporate the CLIP model, and the experimental results show that the proposed method achieves a significant performance improvement. However, the paper lacks a direct comparison with the C3PR method and does not provide sufficient analysis of how each proposed module contributes to the performance gains over C3PR.
2. The proposed method lacks sufficient novelty. The ideas of cross-modal interaction and sampling based on feature statistical information have already been explored in fields outside 3D FSCIL. For example, cross-modal interaction has been discussed in [2], and sampling based on sample statistics has been presented in [3]. Simply introducing these existing concepts into this context does not demonstrate clear innovation. The authors should clarify in the motivation section how applying these ideas to the 3D FSCIL domain differs from other fields.
3. The paper lacks a discussion on the differences between randomly saving prototypes and using Gaussian memory. Although the authors analyze the impact of removing Gaussian memory in the ablation study, this experiment alone cannot determine the respective contributions of sampling and storing sample statistics. Therefore, additional experiments are needed to compare random sampling with the preservation of sample statistics.
4. Concerns about the 2D aggregator: The paper lacks implementation details of the 2D aggregator. I have doubts regarding its effectiveness — the 2D aggregator is intended to integrate depth maps from different viewpoints, but the pixel-space meanings of these depth maps are not aligned. How is the validity of this integration ensured? If the method simply applies pooling or weighted summation over corresponding pixels, it may disrupt the semantic consistency among the feature maps. The authors are encouraged to provide more implementation details and include additional experiments to demonstrate that the 2D aggregator does not introduce negative effects.
5. In the overall framework diagram, the symbols f and F in the FSCIL Architecture are not labeled with their corresponding 3D and 2D spatial dimensions. It is recommended to specify the full dimensions to avoid potential misunderstandings.

[1] Cheraghian A, Hayder Z, Ramasinghe S, et al. Canonical shape projection is all you need for 3d few-shot class incremental learning[C]//European Conference on Computer Vision. Cham: Springer Nature Switzerland, 2024: 36-53.
[2] Li L H, Zhang P, Zhang H, et al. Grounded language-image pre-training[C]//Proceedings of the IEEE/CVF conference on computer vision and pattern recognition. 2022: 10965-10975.
[3] Zhu F, Zhang X Y, Wang C, et al. Prototype augmentation and self-supervision for incremental learning[C]//Proceedings of the IEEE/CVF conference on computer vision and pattern recognition. 2021: 5871-5880.

**Questions:**

Please refer to the points listed in the Weaknesses.

---

### Official Review · Reviewer_cQUP · 2025-10-30

**Soundness:** 1
**Presentation:** 1
**Contribution:** 2
**Rating:** 2
**Confidence:** 5

**Summary:**

This paper proposes GAUSS-Fusion, a multimodal framework for 3D Few-Shot Class-Incremental Learning (FSCIL). The method integrates 2D renderings and 3D point clouds via bidirectional cross-modal attention (SGA and GGA), employs a category-aware viewpoint selection strategy, and introduces a Gaussian Memory to mitigate catastrophic forgetting by storing and replaying feature distributions of old classes. Experiments on multiple 3D datasets demonstrate improvements in relative accuracy dropping rate under both within-dataset and cross-dataset FSCIL settings.

**Strengths:**

1.	The problem setting (3D FSCIL) is practically relevant, especially given the scarcity of real-world 3D annotations.
2.	The paper covers commonly used benchmark datasets in the field and evaluates both cross-dataset and within-dataset incremental learning scenarios.

**Weaknesses:**

**1. Unclear Motivation and Insufficient Innovation**

The paper has obvious flaws in the motivation behind its core design and the innovation of its method. First, it fails to adequately explain why it chooses to transfer the 2D CLIP model to the 3D domain, rather than directly using 3D foundation models (such as OpenShape and UNI3D) that have been widely applied in 3D FSCIL. It should be noted that 3D foundation models can natively capture 3D geometric information, while relying on 2D CLIP will instead introduce cross-modal domain gaps. The motivation for this core design choice has not been clearly elaborated.

Second, the method has limited innovation. The category-aware viewpoint selection strategy is highly similar to the core ideas of the existing method C3PR, lacking substantial improvements that can constitute core innovation. In addition, the Gaussian Memory module is a common solution in the field of continual learning. Overall, the innovation of the method is obviously insufficient.

**2. Inadequate Experimental Evaluation and Doubtful Result Credibility**

The evaluation system in the experimental part is incomplete, and the credibility of the results is controversial. On the one hand, the evaluation metrics are incomplete. The paper only reports the relative accuracy dropping rate, which focuses on the performance retention of old classes, but fails to evaluate the recognition accuracy of new classes. Without the accuracy data of new classes, it is difficult to effectively support the core claim of "mitigating forgetting" proposed in the paper.

More critically, there is a problem of unverifiable experimental results. The data of the CO3D dataset in Table 1 is disorganized, and the Δ value calculated according to the formula and existing data provided in the paper is inconsistent with the reported Δ value. This undoubtedly undermines the credibility of the overall experimental results. The results in Table 2 suffer from the same issue.

**3. About Paper Writing and Presentation Quality**

The paper has many problems with writing standards and content presentation. For example, the method name is sometimes written as "gauss-fusion" and sometimes as "GAUSS-Fusion"; the dataset name "Co3D" and "CO3D" are used interchangeably; the arrow notation is also inconsistent, with "->" and "→" appearing alternately. Even the format of the same method name is inconsistent, sometimes in italics and sometimes not.

The font in Figure 1 is too small, making the content difficult to identify, and some text labels (such as "Text Prompt") lack necessary punctuation marks. The table format is also non-standard. Some table titles do not have end punctuation, and there is a spelling error in Table 5, where "without Attention" is incorrectly written as "without Attetion".

After the first definition of FSCIL, it is still unnecessarily labeled with the abbreviation repeatedly.

**Questions:**

1.	Why does the paper choose to transfer the 2D CLIP model to the 3D FSCIL task instead of directly adopting or building on those 3D foundation models? In addition, why are these 3D foundation models not included in the experimental comparison [1]?
2.	As mentioned earlier, the paper lacks comparisons with the latest SOTA methods.
3.	In terms of experimental setup and evaluation, several details need to be further clarified: first, please supplement the description of the number of classes and specific classes included in each incremental learning stage; second, what is the basis for choosing 1024 points as the point cloud input in the paper; finally, in addition to the relative accuracy dropping rate, can you provide other indicators that can more accurately reflect the degree of forgetting?
4.	The results of the CO3D dataset in Tables 1 and 2 contain unrecognizable content. Please check and clarify this data. At the same time, the Δ value calculated based on the reported initial accuracy and final accuracy is inconsistent with the Δ value reported in the table. Please verify and explain the reason for this difference.
5.	This paper contains numerous writing inconsistencies (as outlined in the “Weaknesses” section).

[1] Foundation Model-Powered 3D Few-Shot Class Incremental Learning via Training-free Adaptor, ACCV2024.

---

### Official Review · Reviewer_4ijA · 2025-11-01

**Soundness:** 2
**Presentation:** 2
**Contribution:** 2
**Rating:** 4
**Confidence:** 4

**Summary:**

Against the key challenges of 3D Few-Shot Class-Incremental Learning, this paper introduces GAUSS-Fusion. The framework’s design focuses on building robust multi-modal representations and mitigating forgetting: it uses bidirectional cross-modal attention to link 2D renderings and 3D points, adopts a category-aware strategy to select discriminative viewpoints for 2D projection, and proposes a Gaussian Memory module that stores the statistical distributions (mean and variance) of old class features for generative replay.

**Strengths:**

The paper features comprehensive experimental coverage, addressing multiple practical 3D FSCIL scenarios that reflect real-world 3D data challenges. It utilizes four representative 3D point cloud datasets, including two real-world ones and two synthetic ones, to avoid bias toward idealized data; designs cross-dataset experiments to simulate the practical scenario of pre-training on synthetic data and fine-tuning on real scans, a critical setting in 3D FSCIL ; and conducts robustness tests on point shift noise (with σ=0.001 and σ=0.01) and random point drop to mimic common sensor or environmental issues in 3D data acquisition.

**Weaknesses:**

The paper lacks sufficient originality, as core modules are not novel in isolation—cross-modal fusion for 3D and distribution-based memory have been widely explored, with limited analysis of how GAUSS-Fusion differs from these existing methods. Efficiency is not discussed, with no metrics for computational overhead, parameter count, or memory usage. Experimental details are incomplete: key parameters lack justification, and training details like augmentation magnitudes are unspecified, harming reproducibility. Finally, writing and formatting issues reduce professionalism and readability.

**Questions:**

In the "relative accuracy dropping" metric, does "the first incremental tasks" specifically refer to the base training phase or the first incremental phase?

The Related Works section does not include some of the latest works. Additionally, among the comparative methods, C3PR is already a paper a 2024 conference, and the study may be missing some recent methods.

The paper has some unreasonable paragraph breaks and some critical citations are missing.

The paper claims GAUSS-Fusion’s innovation lies in combining cross-modal attention and Gaussian Memory, but it fails to detail how this combination differs from existing methods. What specific technical differences make GAUSS-Fusion’s design innovative, beyond simply assembling existing module ideas?

No efficiency metrics are provided. For incremental learning models that require deployment in resource-constrained scenarios, how can the practicality of GAUSS-Fusion be verified without such metrics?

---

### Note · Authors · 2025-11-13

I have read and agree with the venue's withdrawal policy on behalf of myself and my co-authors.